# Abdominal Organs and Pan-Cancer Segmentation based on Self-supervised Pre-training and Self-training

He Li[1], Meng Han[1], and Guotai Wang[1,2]

[1] School of Mechanical and Electrical Engineering, University of Electronic Science and Technology of China, Chengdu, China
[2] Shanghai Artificial Intelligence Laboratory, Shanghai, China
{guotai.wang}@uestc.edu.cn

**Abstract.** Despite the effective progress in automatic abdominal multi-organ segmentation methods based on deep learning, there are still few studies on general models for abdominal organ and pan-cancer segmentation. Additionally, the manual annotation of organs and tumors from CT scans is a time-consuming and labor-intensive process. To deal with these problems, an efficient two-stage framework combining self-supervised pre-training and self-training is proposed. Specifically, in the first stage, we adopt the Model Genesis method for image reconstruction to promote the model to learn effective anatomical representation information, thereby improving the model's perception of anatomical structures in downstream segmentation tasks and generating high-quality tumor pseudo-labels. Afterward, we fuse partial organ fine-standard of labeled data with pseudo-labels to improve the organ labeling quality. In the second stage, we overlay the generated tumor pseudo-labels onto the corresponding regions of the organ pseudo-labels, and the final pseudo-label images are used to train the nnU-Net model for efficient inference. The proposed method has been evaluated on the FLARE2023 validation cased, and get a relatively good segmentation performance. The average DSC and NSD for organs are 91.51% and 95.52%, respectively. For tumors, the average DSC is 43.47%, and the average NSD is 33.81%. In addition, the average running time and area under the GPU memory-time curve are 85.4 s and 246157.2 MB, respectively. On the test set, we achieved average organ and tumor DSC of 92.17% and 54.99%, respectively, and average inference time of 95.83 s. Our code is publicly available at https://github.com/lihe-CV/HiLab_FLARE23

**Keywords:** Semi-supervised learning · Self-supervised learning · Pseudo labels.

## 1 Introduction

Abdominal organ and tumor segmentation is a critically important task in abdominal disease diagnosis, cancer treatment, and radiation therapy planning [11].

The abdomen is a common site for the occurrence of cancer, and accurate segmentation results can provide valuable information for clinical diagnosis and surgical planning, like the size and location of organs and tumors, the spatial relationship of multiple organs, etc. In recent years, deep learning-based methods have been widely used for automatic segmentation of organs and tumors [16]. However, these methods heavily rely on a large amount of annotated data for training purposes. In past clinical practice, segmentation labels for organs and tumors was usually performed manually by radiologists. It is time-consuming and labor-intensive. Thus, it is often challenging to obtain a large number of labeled cases. In light of this situation, semi-supervised semantic segmentation aims to utilize limited labeled data and abundant unlabeled data for model training. It addresses the issue of label scarcity by exploring valuable information from the unlabeled data.

FLARE [13] is an international challenge focusing on abdominal scene segmentation. Compared with FLARE22, the challenge for FLARE23 adds the pancancer segmentation task and provides only partial organ segmentation labels for the labeled data in semi-supervised segmentation. The organizer of FLARE23 provided the largest abdomen CT dataset, including 4000 3D CT scans from 30+ medical centers. 2200 cases have partial labels and 1800 cases are unlabeled. For the task scenario combining semi-supervised and partial-label segmentation, the main solutions can be divided into two types: (1) consistency-regularization-based methods [3]. (2) pseudo-label-based methods [10,19]. Since the organizer invited the FLARE22 champion team to generate pseudo labels for FLARE23 data. We choose the pseudo-labeling-based approach and integrate it with the nnU-Net framework [9] to train organ segmentation model. However, due to the uncertainty in tumor shape, size, and location, as well as the scarcity of tumor labels, we attempt to incorporate self-supervised strategies to learn effective representation information from images, thereby enhancing the model's perception of tumor category.

In this work, we propose a two-stage training framework that combines self-supervised and semi-supervised learning to generate high-quality pseudo-labels and improve the segmentation performance of the model, respectively. Specifically, in the first stage, we employ the Model Genesis method [23] for image reconstruction to learn effective anatomical representation information. From 2200 labeled images, 735 tumor-containing images and corresponding labels were further selected, and the pre-trained model was transferred to the tumor segmentation task to generate high-quality pseudo-labels for 3265 tumor-free labeled data [2]. For the pseudo-label generation of organs, we simply fused the pseudo-labels provided by the organizer with partial organ segmentation annotations, and achieved good segmentation results. In the second stage, we overlay the generated tumor pseudo-labels onto the corresponding regions of the organ pseudo-labels, and the final pseudo-labeled images are used to train nnU-Net model [9] for inference.

In summary, we make the following three contributions:

- We design a two-stage training framework based on nnU-Net to generate high-quality pseudo-labels and improve the segmentation performance of the model.
- We adopt self-supervised learning strategy to learn anatomical representation information, enabling the model to generate high-quality pseudo-labels.
- We optimize the organ segmentation task by fusing pseudo-labels and partial organ segmentation annotations. Models trained with our fused labels perform better.

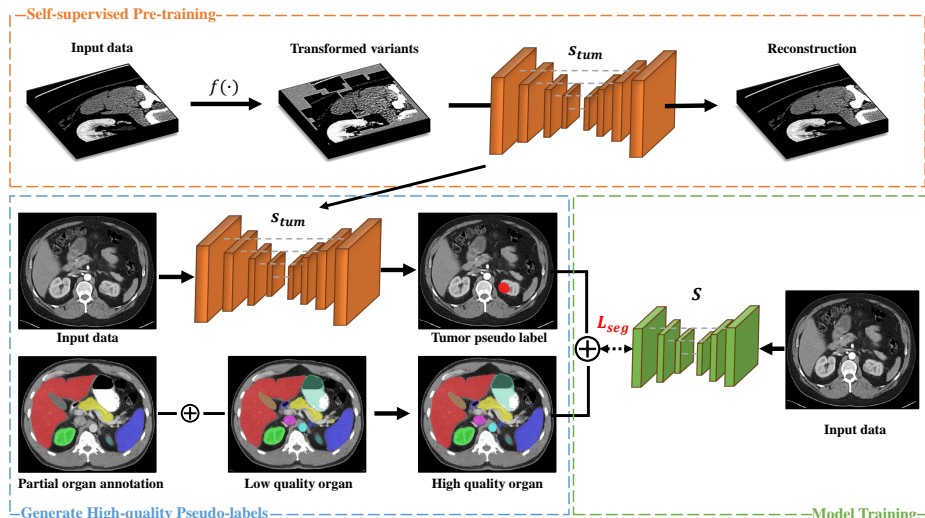

**Fig. 1.** Overview of our proposed framework.

## 2   Method

To deal with a training dataset with partial labels on a small part of images, we propose a two-stage training framework that combines self-supervised and semi-supervised learning, as shown in Figure 1. We adopt self-supervised learning and image fusion strategies to generate high-quality pseudo-labels. The self-training [10] is adopted for semi-supervised semantic segmentation. The detailed description of this framework is as follows.

### 2.1   Preprocessing

The preprocessing strategies for labeled data and pseudo-labeled data in the two-stage segmentation framework are as follows:

**Table 1.** Comparison of different segmentation models. The order of axes of input patch size and spacing is (z,y,x).

| Settings | Default | Tumor | Organ&Pan-cancer |
|---|---|---|---|
| convolution kernel sizes | (1, 3, 3) | (3, 3, 3) | (1, 3, 3) |
| step size for sliding window | 0.5 | 0.5 | 1 |
| input patch size | (64×160×192) | (112×160×160) | (56×160×192) |
| input spacing | (2.0, 0.8, 0.8) | (1.8, 1.8, 1.8) | (2.5, 0.8, 0.8) |

- Image cropping: Crop the bounding box of the image to the non-zero area, thereby reducing the image size and improving computational efficiency.
- We adopt image resampling to ensure that the actual physical space of each voxel is consistent across different image data.
- We applied z-score normalization based on the mean and standard deviation of foreground intensity values across the training set.
- The detailed configurations and the comparison with default nnU-Net are listed in Table 1.

### 2.2 Generate High-quality Pseudo-labels

We employ pseudo-label generation as a simple and effective method to utilize unlabeled data for model training. Specifically, we make full use of the pseudo-labels of abdominal organs provided by the organizer. To improve the labeling quality, we fuse partial organ fine-standard of labeled data with pseudo-labels. However, tumor category is difficult to segment due to the uncertainty of tumor shape, size and location, as well as the scarcity of tumor labels. We utilize self-supervised learning strategy to facilitate tumor segmentation model $S_{tum}$ to understand local and global features, thereby boosting the model's awareness of tumor category and generating high-quality tumor pseudo-labels.

**Self-supervised Pre-training.** Model Genesis [23] learns from scratch on un-labeled images with the goal of learning a universal visual representation that can be generalized and transferred across diseases, organs, and modalities. In order to improve the model's transfer and perception capabilities for tumor category, we use similar self-supervised training strategies as the Model Genesis [23] to pre-train $S_{tum}$ with the provided FLARE23 dataset. Throughout the pre-training process, $S_{tum}$ reconstructs the original patches according to the augmented variants, thereby learning anatomical representation information of 3D abdominal CT images. The generation process of augmented variants is shown in Figure 2.

Specifically, four transformations are randomly combined and applied to the original patch to generate augmented variants. The transformations include: 1) Non-linear transformation. By integrating Bézier Curve [17] to assign a uniquely determined value to each pixel, to encourage self-supervision focusing on the information of image appearance, shape and intensity distribution. 2) Local pixel shuffling. By sampling a window smaller than the model's receptive field in the

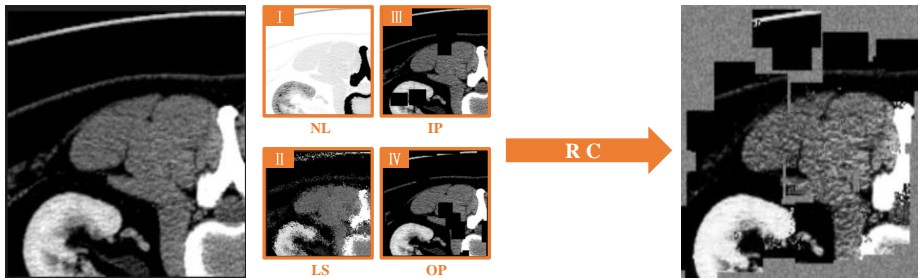

**Fig. 2.** The transformations made to the original patch during the pre-training. I:Nonlinear transformation, II:local pixels shuffling, III: in-painting, IV: out-painting. (RC: random combine.)

patch and rearranging the internal pixels, to encourage model learning the local texture and boundary of the image. 3) Out-painting and In-painting. By blending windows of different sizes to create a complex shape. Out-painting sets the outer pixels of the shape to random values, while the inner pixels retain their original intensities. In-painting is the opposite.

Then, the pre-training model $S_{tum}$ will learn the anatomical representation information by reconstructing the original patch. The mean squared error (MSE) loss is used for training $S_{tum}$ by minimizing a reconstruction error $\mathcal{L}_{rec}$:

$$\mathcal{L}_{rec} = \frac{1}{N} \sum_{i=1}^{N} \left| X_i - \hat{X}_i \right| \tag{1}$$

where $i$ is the voxel index, $N$ is the number of the voxels, $X_i$ is original patch and $\hat{X}_i$ is the prediction of the model. Finally, we screened out 735 tumor-containing images and corresponding labels from 2200 labeled data, and transferred the pre-trained model $S_{tum}$ to the tumor segmentation task. We adopt an average of cross-entropy loss and Dice loss to supervise the tumor segmentation model:

$$\mathcal{L}_{seg} = \frac{1}{2N^t} \sum_{i=1}^{N^t} \left( \mathcal{L}_{Dice}(p_i, y_i) + \mathcal{L}_{ce}(p_i, y_i) \right) \tag{2}$$

where $y_i$ is the tumor label, $N^t$ is the number of training images and $p_i$ is the prediction of the model $S_{tum}$.

**Label Fusion.** Due to the organizer invited FLARE22 champion team to use its docker to generate pseudo labels for FLARE23 data. Therefore, we adopt a simple but effective label fusion strategy. Specifically, we achieve high-quality fusion by replacing the corresponding organ regions in the pseudo-labels with accurately annotated organ parts from the labeled data.

$$\hat{y} = y_p \oplus y_q \tag{3}$$

where $y_p$ is ground truth and $y_q$ is pseudo label. At the same time, the unlabeled data retains the corresponding pseudo-labels as supervision signals.

### 2.3   Model Training and Inference

We adopt similar label fusion strategy to the high-quality organ and tumor pseudo-labels obtained in the first stage, generating a dataset $\mathcal{D} = \{x_i, y_i\}_{i=1}^{N}$ for training organ and pan-cancer segmentation model $S$. In addition, in order to improve the inference efficiency of the model $S$, we try using small patch size as in Table 1 to increase the training and inference speed of each patch and reduce GPU memory. Finally, the segmentation model $S$ learns from organ and pan-cancer data by minimizing a supervised loss function:

$$\mathcal{L}_{seg} = \frac{1}{2N^d} \sum_{j=1}^{N^d} \left( \mathcal{L}_{Dice}(p_j, y_j) + \mathcal{L}_{ce}(p_j, y_j) \right) \tag{4}$$

where $y_j$ is the organ and pan-cancer label, $N^d$ is the number of training images and $p_j$ is the prediction of the model $S$.

Due to the high resolution of 3D medical images, nnU-Net [9] adopts the sliding-window strategy for inference. However, this strategy significantly consumes the time and space complexity. Therefore, we set the step-size to 1 during inference to effectively improve inference speed and reduce resource consumption while ensuring accuracy.

### 2.4   Post-processing

A connected component analysis of segmentation mask is applied on the outputs to remove small connected areas. And then the results are resampled back to original spacing for the convenience of the following evaluation.

## 3   Experiments

### 3.1   Dataset and evaluation measures

The FLARE 2023 challenge is an extension of the FLARE 2021-2022 [13][14], aiming to promote the development of foundation models in abdominal disease analysis. The segmentation targets cover 13 organs and various abdominal lesions. The training dataset is curated from more than 30 medical centers under the license permission, including TCIA [4], LiTS [1], MSD [20], KiTS [7,8], autoPET [6,5], TotalSegmentator [21], and AbdomenCT-1K [15]. The training set includes 4000 abdomen CT scans where 2200 CT scans with partial labels and 1800 CT scans without labels. The validation and testing sets include 100 and 400 CT scans, respectively, which cover various abdominal cancer types, such as liver cancer, kidney cancer, pancreas cancer, colon cancer, gastric cancer, and so on. The organ annotation process used ITK-SNAP [22], nnU-Net [9], and MedSAM [12].

The evaluation metrics encompass two accuracy measures—Dice Similarity Coefficient (DSC) and Normalized Surface Dice (NSD)—alongside two efficiency measures—running time and area under the GPU memory-time curve. These

metrics collectively contribute to the ranking computation. Furthermore, the running time and GPU memory consumption are considered within tolerances of 15 seconds and 4 GB, respectively.

### 3.2   Implementation details

**Environment settings** The development environments and requirements are presented in Table 2.

**Table 2.** Development environments and requirements.

| | |
|---|---|
| System vision | Ubuntu 18.04.5 LTS |
| CPU | Intel(R) Xeon(R) Gold 6248 CPU@2.50GHz |
| RAM | 16×4GB; 2.67MT/s |
| GPU (number and type) | One NVIDIA V100 32G |
| CUDA version | 11.0 |
| Programming language | Python 3.10.8 |
| Deep learning framework | torch 2.0.0, torchvision 0.15.1 |
| Specific dependencies | nnU-Net 2.1.1 |
| Code | https://github.com/lihe-CV/HiLab_FLARE23 |

**Training protocols** The training protocols of $S_{tum}$ and $S$ are shown in Table 3 and 4 respectively. During the training process, we dynamically adopt elastic deformation, rotation, random cropping, Gaussian noise transformation, Gamma transformation, contrast transformation, morphological transformation and other data enhancement strategies. In addition, we applied mirror test time data augmentation during inference.

## 4   Results and discussion

### 4.1   Quantitative results on validation set

Quantitative result is illustrated in Table 5, it can be observed that the two-stage framework can achieve very promising segmentation results for large regional organs, such as liver, spleen, kidney, stomach, etc. However, the segmentation of small and structurally complex organs such as the duodenum, esophagus, and adrenal glands remains challenging in comparison. Moreover, the strong uncertainty in tumor shapes, sizes, and locations in the pan-cancer segmentation task added to the FLARE23 challenge makes the segmentation task extremely challenging. Indeed, there is a problem of missing in the segmentation results, particularly for small tumors, where the segmentation model fails to predict their presence.

**Table 3.** Training protocols for tumor segmentation model $S_{tum}$.

| | |
|---|---|
| Network initialization | "He" normal initialization |
| Batch size | 2 |
| Patch size | 112×160×160 |
| Total epochs | 2000 |
| Step size | 1 |
| Optimizer | SGD with nesterov momentum ($\mu = 0.99$) |
| Initial learning rate (lr) | 0.01 |
| Lr decay schedule | Poly learning rate policy: $(1 - epoch/2000)^{0.9}$ |
| Training time | 132.5 hours |
| Loss function | Dice loss and cross entropy loss |
| Number of model parameters 88.21M | |
| Number of flops | 913.4G |
| $CO_2$eq | 41.05 Kg |

**Table 4.** Training protocols for organ and pan-cancer segmentation model $S$.

| | |
|---|---|
| Network initialization | "He" normal initialization |
| Batch size | 2 |
| Patch size | 56×160×192 |
| Total epochs | 1000 |
| Optimizer | SGD with nesterov momentum ($\mu = 0.99$) |
| Initial learning rate (lr) | 0.01 |
| Lr decay schedule | Poly learning rate policy: $(1 - epoch/1000)^{0.9}$ |
| Training time | 41.5 hours |
| Loss function | Dice loss and cross entropy loss |
| Number of model parameters 71.02M | |
| Number of flops | 727.76G |
| $CO_2$eq | 35.02 Kg |

**Table 5.** Quantitative results of validation set in terms of DSC and NSD. (Public Validation: the performance on the 50 validation cases with ground truth. Online Validation: the leaderboard results. Testing: the performance on the testing cases.)

| Target | Public Validation | | Online Validation | | Testing | |
|---|---|---|---|---|---|---|
| | DSC(%) | NSD(%) | DSC(%) | NSD(%) | DSC(%) | NSD (%) |
| Liver | 98.43 ± 0.0102 | 98.91 ± 0.0231 | 98.30 | 98.77 | 96.50 | 96.79 |
| Right Kidney | 93.25 ± 12.12 | 94.13 ± 11.83 | 93.47 | 93.47 | 93.55 | 93.14 |
| Spleen | 96.52 ± 11.39 | 96.74 ± 13.23 | 95.82 | 96.53 | 96.42 | 96.65 |
| Pancreas | 85.43 ± 10.89 | 95.35 ± 10.06 | 86.84 | 96.54 | 91.36 | 97.30 |
| Aorta | 97.45 ± 1.82 | 99.41 ± 2.55 | 97.68 | 99.29 | 97.58 | 98.91 |
| Inferior vena cava | 93.02 ± 7.29 | 92.66 ± 6.91 | 93.21 | 93.87 | 93.60 | 95.05 |
| Right adrenal gland | 88.94 ± 9.21 | 97.88 ± 10.23 | 89.09 | 97.89 | 87.95 | 95.99 |
| Left adrenal gland | 87.89 ± 9.63 | 96.17 ± 8.31 | 87.58 | 95.61 | 89.58 | 96.68 |
| Gallbladder | 88.95 ± 20.48 | 90.79 ± 21.38 | 89.79 | 90.54 | 85.40 | 87.20 |
| Esophagus | 84.22 ± 10.06 | 93.76 ± 10.18 | 84.19 | 93.30 | 89.24 | 96.28 |
| Stomach | 95.01 ± 3.71 | 97.90 ± 6.65 | 94.66 | 97.18 | 94.84 | 96.65 |
| Duodenum | 83.74 ± 10.01 | 94.98 ± 8.24 | 84.53 | 95.21 | 88.82 | 96.32 |
| Left kidney | 94.81 ± 12.69 | 93.31 ± 12.80 | 94.44 | 93.60 | 92.89 | 92.48 |
| Tumor | 43.86 ± 25.98 | 33.27 ± 23.56 | 43.47 | 33.81 | 54.99 | 42.45 |
| Average(Organ) | 91.62 ± 9.73 | 95.38 ± 14.70 | 91.51 | 95.52 | 92.17 | 95.44 |

Then, Table 6 and Table 7 showed the Dice and NSD metrics calculated on the validation set. Evidently, compared with models trained using labeled data with only partial organ segmentation annotations, training the model using the Label Fusion strategy can significantly improve segmentation performance. Moreover, the introduction of self-supervised pretraining strategy has significantly improved the performance of the two-stage framework on tumor classes, as evidenced by the achieved Dice Similarity Coefficient (DSC) of 43.47%.

**Table 6.** Ablation study of Dice(%) metrics on validation set. (BaseLine: Training nnU-Net with labeled images only. LF: Label Fusion. SP: Self-supervised Pre-training.)

| Methods | Liver | RK | Spleen | Pancreas | Aorta | IVC | RAG | LAG |
|---|---|---|---|---|---|---|---|---|
| Baseline | 97.58 | 92.71 | 94.96 | 85.94 | 97.01 | 91.29 | 82.32 | 83.69 |
| Baseline+LF | **98.46** | **95.98** | **97.10** | 86.72 | 97.51 | **93.34** | 88.46 | **88.77** |
| Baseline+LF+SP | 98.30 | 93.47 | 95.82 | **86.84** | **97.68** | 93.21 | **89.09** | 87.58 |

| Methods | GBD | EPG | Stomach | Duodenum | LK | Average | Tumor |
|---|---|---|---|---|---|---|---|
| Baseline | 85.11 | **85.69** | 91.24 | 80.65 | 93.29 | 89.38 | 33.06 |
| Baseline+LF | 88.34 | 84.35 | 94.35 | **84.64** | **95.22** | **91.68** | 37.52 |
| Baseline+LF+SP | **89.79** | 84.19 | **94.66** | 84.53 | 94.44 | 91.51 | **43.47** |

Finally, we quantitatively evaluated the segmentation efficiency of the model, as shown in Table 8. It can be found that the three evaluation metrics show an increasing trend as the input instances grow larger. Although the inference time is mostly within 60 seconds, the proportion of inference times below 15 seconds

**Table 7.** Ablation study of NSD(%) metrics on validation set. (BaseLine: Training nnU-Net with labeled images only. LF: Label Fusion. SP: Self-supervised Pre-training.)

| Methods | Liver | RK | Spleen | Pancreas | Aorta | IVC | RAG | LAG |
|---|---|---|---|---|---|---|---|---|
| Baseline | 97.83 | 92.12 | 96.12 | 95.24 | 98.35 | 91.67 | 92.29 | 92.41 |
| Baseline+LF | **99.05** | **96.16** | **98.25** | 96.46 | 99.25 | **94.01** | 97.69 | **96.85** |
| Baseline+LF+SP | 98.77 | 93.47 | 96.53 | **96.54** | **99.29** | 93.87 | **97.89** | 95.61 |

| Methods | GBD | EPG | Stomach | Duodenum | LK | Average | Tumor |
|---|---|---|---|---|---|---|---|
| Baseline | 85.74 | 92.98 | 94.62 | 91.33 | 92.01 | 93.91 | 22.07 |
| Baseline+LF | 89.84 | **93.55** | 96.92 | **95.33** | **95.08** | **96.07** | 28.52 |
| Baseline+LF+SP | **90.54** | 93.30 | **97.18** | 95.21 | 93.60 | 95.52 | **33.81** |

is relatively low. Therefore, further optimization is needed in terms of model inference efficiency to strive for achieving clinical usability standards.

**Table 8.** Quantitative evaluation of segmentation efficiency in terms of the running them and GPU memory consumption.(Total GPU: the area under GPU Memory-Time curve. Evaluation GPU platform: NVIDIA QUADRO RTX5000 (16G).)

| Case ID | Image Size | Running Time (s) | Max GPU (MB) | Total GPU (MB) |
|---|---|---|---|---|
| 0001 | (512, 512, 55) | 8.47 | 4266 | 17433 |
| 0051 | (512, 512, 100) | 11.12 | 5290 | 34759 |
| 0017 | (512, 512, 150) | 20.29 | 5526 | 70528 |
| 0019 | (512, 512, 215) | 23.62 | 4722 | 64923 |
| 0099 | (512, 512, 334) | 30.20 | 5282 | 96292 |
| 0063 | (512, 512, 448) | 42.10 | 5506 | 144037 |
| 0048 | (512, 512, 499) | 49.15 | 5420 | 150701 |
| 0029 | (512, 512, 554) | 97.21 | 6142 | 247289 |

### 4.2   Qualitative results on validation set

Figure 3 displays the qualitative results on the validation set. The first and second rows depict relatively easy segmentation cases, while the third and fourth rows showcase challenging segmentation cases. It can be observed that in the first and second rows, the organ boundaries are clear, there is good contrast, and there are no complex tumor lesions within the organs. Compared with well-segmented instances, challenging instances often have complex tumor lesions (row 3) and noise (row 4), which bring difficulties to accurate segmentation of organs and pan-cancer.

### 4.3   Segmentation efficiency results on validation set

We combine efficient inference schemes to build nnU-Net [9] as the final submitted Docker image. The average running time per instance during the inference

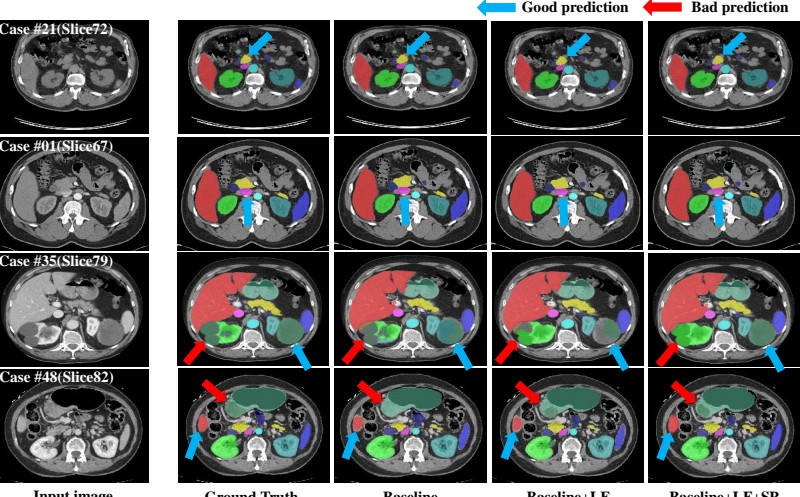

**Fig. 3.** Qualitative evaluation of model performance on validation set. Row 1 and 2: Well-segmented examples. Row 3 and 4: challenging examples.

phase is 85.4 seconds, and average used GPU memory is 2352 MB. The area under the GPU memory-time curve is 246157.2 MB, and the area under CPU utilization-time curve is 2973.

### 4.4 Results on final testing set

Table 5 show the detailed evaluation metrics of our method in the final testing set. It can be observed that the two-stage framework achieved average DSC scores of 92.17% for organs and 54.99% for lesions, along with NSD scores averaging 95.44% for organs and 42.45% for lesions. Additionally, the average running time was 95.83 seconds, and the area under the GPU memory-time curve was 227770 MB.

### 4.5 Limitation and future work

While ensuring accuracy, we can explore the use of the following advanced processing strategies to speed up inference and reduce resource consumption:

- Model Pruning. Identify and remove redundant or less important model parameters, reducing the model size and improving inference speed without significant loss in accuracy.
- Model Quantization. Convert the model from floating-point precision to lower-precision fixed-point representation, reducing memory usage and improving inference speed.
- Filter Data Augmentation. Select specific data enhancement strategies based on organ and tumor characteristics to prevent redundancy.

## 5   Conclusion

In this work, we propose a two-stage training framework that combines self-supervised and semi-supervised learning to efficiently perform training and inference on organ and pan-cancer segmentation tasks. Experiments show that our method achieves good segmentation performance. In the future, we hope to optimize the model framework to further improve the segmentation accuracy of difficult tumor samples, improve inference speed and reduce resource consumption.

**Acknowledgements** The authors of this paper declare that the segmentation method they implemented for participation in the FLARE 2023 challenge has not used any pre-trained models nor additional datasets other than those provided by the organizers. The proposed solution is fully automatic without any manual intervention. We thank all the data owners for making the CT scans publicly available and CodaLab [18] for hosting the challenge platform.

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

**Table 9.** Checklist Table. Please fill out this checklist table in the answer column.

| Requirements | Answer |
|---|---|
| A meaningful title | Yes |
| The number of authors ($\leq 6$) | 3 |
| Author affiliations and ORCID | Yes |
| Corresponding author email is presented | Yes |
| Validation scores are presented in the abstract | Yes |
| Introduction includes at least three parts: background, related work, and motivation | Yes |
| A pipeline/network figure is provided | 1 |
| Pre-processing | 3 |
| Strategies to use the partial label | 5 |
| Strategies to use the unlabeled images. | 4 |
| Strategies to improve model inference | 5 |
| Post-processing | 6 |
| Dataset and evaluation metric section is presented | 6 |
| Environment setting table is provided | 7 |
| Training protocol table is provided | 7 |
| Ablation study | 9 |
| Efficiency evaluation results are provided | 10 |
| Visualized segmentation example is provided | 11 |
| Limitation and future work are presented | Yes |
| Reference format is consistent. | Yes |