# OpenReview forum: "Abdominal Organs and Pan-Cancer Segmentation based on Self-supervised Pre-training and Self-training"
_MICCAI.org/2023/FLARE — Submitted to FLARE 2023_

### Official Review · Reviewer_13Kx · 2023-09-24
**Excellent work with slight issues**

**Rating:** 8
**Confidence:** 5

**Review:**

Summary:
The paper presents an efficient two-stage framework that combines self-supervised pre-training and self-training to address the FLARE23 challenge, achieving impressive performance. The ablation study demonstrates that self-supervised pre-training significantly enhances cancer segmentation.

Pros:
The paper is well-written, offering a clear and detailed introduction to the methodology and a comprehensive presentation and analysis of the results.

Cons:
1. In the Introduction, it is stated that 735 tumor-containing images were selected to train the tumor segmentation model. However, there are more than 735 images in the 2200 labeled data with cancer labels. An explanation of how the 735 images were chosen would be helpful.
2. Consider increasing the text size in Figure 1 for better readability.
3. In Figure 3, it would be beneficial to specify the particular slices presented and provide their respective dice scores. Additionally, highlighting areas where segmentation is less accurate would improve the clarity of the visualization.

---

> ### Author Response · Authors · 2023-11-14
> **Reply towards reviewer 13Kx's comments**
>
> Thank you very much for your constructive comments on this article. We have carefully considered your suggestions and will respond to them one by one:
> 1. In the Introduction, it is stated that 735 tumor-containing images were selected to train the tumor segmentation model. However, there are more than 735 images in the 2200 labeled data with cancer labels. An explanation of how the 735 images were chosen would be helpful.
>
> We acknowledge that this problem resulted from an oversight in our screening of the oncology data. When using nib to read data, some of the tumor class values were floating point numbers of 14.00005. I did not convert them into integers at that time, so I did not filter out all the tumor data. I personally infer that using all tumor data for model training will further improve the effect.
>
> 2. Consider increasing the text size in Figure 1 for better readability.
>
> Thank you for your suggestion. The revised manuscript will increase the text size in Figure 1.
>
> 3. In Figure 3, it would be beneficial to specify the particular slices presented and provide their respective dice scores.Additionally, highlighting areas where segmentation is less accurate would improve the clarity of the visualization.
>
> Thank you for your suggestion. The revised manuscript would specify the specific slices presented and provide their respective dice scores, while also highlighting areas where the segmentation was less accurate.

---

### Official Review · Reviewer_yeUR · 2023-09-30
**Well-structured article with minor errors**

**Rating:** 8
**Confidence:** 4

**Review:**

Pros:
1. Well-structured articles.
2. The proposed method accurately and efficiently segments abdominal organs and tumors. In online validation, the DSC values for organs and tumor segmentation are 91.51% and 43.47%, respectively, and the NSD values are 95.52% and 33.81%, respectively. The inference time was an average of 15.4 seconds, and the area under the GPU memory time curve was an average of 24615.72 MB.

Cons:
1. Abstract: Please add information about the average running time and area under the GPU memory-time curve.
2. It would be highly beneficial if the authors could provide open-source code.
3. Page 5: The 2200-labeled data contains 1497 tumor-labeled data, and only 735 are used in the paper, so why not use all of the data containing tumors?
4. Section 2.4: Is the post-processing operation for all organs and tumors?
5. Section 3.1: "aiming to aim to" should be "aiming to".
6. Section 4.3: Please add "MB" after "24615.72".
7. Table 5, 6, 7 and Fig.3 exceed the default layout.

---

> ### Author Response · Authors · 2023-11-14
> **Reply towards reviewer yeUR's comments**
>
> Thank you very much for your constructive comments on this article. We have carefully considered your suggestions and will respond to them one by one:
> 1. Abstract: Please add information about the average running time and area under the GPU memory-time curve.
>
> Thank you for your suggestion. The revised manuscript will add information about the average running time and the area under the GPU memory time curve in the abstract.
>
> 2. It would be highly beneficial if the authors could provide open-source code.
>
> Thank you for your suggestion. We have open-sourced the code to https://github.com/lihe-CV/HiLab_FLARE23, please check.
>
> 3. Page 5: The 2200-labeled data contains 1497 tumor-labeled data, and only 735 are used in the paper, so why not use all of the data containing tumors?
>
> We acknowledge that this problem resulted from an oversight in our screening of the oncology data. When using nib to read data, some of the tumor class values were floating point numbers of 14.00005. I did not convert them into integers at that time, so I did not filter out all the tumor data. I personally infer that using all tumor data for model training will further improve the effect.
>
> 4. Section 2.4: Is the post-processing operation for all organs and tumors?
>
> Yes, post-processing operations are performed on all organs and tumors.
>
> 5. Section 3.1: "aiming to aim to" should be "aiming to".
>
> Thank you for your suggestion. The revised manuscript will adjust it.
>
> 6. Section 4.3: Please add "MB" after "24615.72".
>
> Thank you for your suggestion. The revised manuscript will adjust it.
>
> 7. Table 5, 6, 7 and Fig.3 exceed the default layout.
>
> Thank you for your suggestion. The revised manuscript will adjust it.

---

### Official Review · Reviewer_3YR5 · 2023-10-03
**Good writing, complete structure with minor issues**

**Rating:** 8
**Confidence:** 4

**Review:**

The authors propose an efficient two-stage framework combining self-supervised pre-training and self-training for abdominal organs and pan-cancer segmentation. Good writing and complete structure. There are several minor issues:

1. Tab.5,6,7 and, Fig. 3 is too large and widths beyond the margins.
2. It is recommended to provide a link to the code in Table 2.
3. Results of segmentation efficiency, such as average segmentation time, can be described in the abstract.
4. The improvement strategy of sliding window proposed by last year's champion has greatly improved the segmentation efficiency, and the author can try to further improve the segmentation efficiency.

---

> ### Author Response · Authors · 2023-11-14
> **Reply towards reviewer 3YR5's comments**
>
> Thank you very much for your constructive comments on this article. We have carefully considered your suggestions and will respond to them one by one:
> 1. Tab.5,6,7 and, Fig. 3 is too large and widths beyond the margins.
>
> Thank you for your suggestion. The revised manuscript will adjust it.
>
> 2. It is recommended to provide a link to the code in Table 2.
>
> Thank you for your suggestion. The revised manuscript will adjust it.
>
> 3. Results of segmentation efficiency, such as average segmentation time, can be described in the abstract.
>
> Thank you for your suggestion. The revised manuscript will add information about the average running time and the area under the GPU memory time curve in the abstract.
>
> 4. The improvement strategy of sliding window proposed by last year's champion has greatly improved the segmentation efficiency, and the author can try to further improve the segmentation efficiency.
>
> Thank you for your suggestion. In the future, I will further adjust the structure to effectively improve the segmentation efficiency.

---

### Official Review · Reviewer_d16s · 2023-10-18
**Well-structured article with minor issues**

**Rating:** 8
**Confidence:** 4

**Review:**

The proposed method achieves accurate and efficient segmentation of abdominal organs and tumors. In online validation, the DSC for organs and tumors segmentation are 91.51% and 43.47%, respectively, and the NSD values are 95.52% and 33.81%, respectively. The inference time was an average of 15.4 seconds and the area under the GPU memory time curve was an average of 24615.72 MB.
There are several minor issues:
1. Abstract: Please add the average running time and area under the GPU memory-time curve.
2. Tables 5, 6, 7, and Fig.3 exceed the default layout.

---

> ### Author Response · Authors · 2023-11-14
> **Reply towards reviewer  d16s's comments**
>
> Thank you very much for your constructive comments on this article. We have carefully considered your suggestions and will respond to them one by one:
> 1. Abstract: Please add the average running time and area under the GPU memory-time curve.
>
> Thank you for your suggestion. The revised manuscript will add information about the average running time and the area under the GPU memory time curve in the abstract.
>
> 2. Tables 5, 6, 7, and Fig.3 exceed the default layout.
>
> Thank you for your suggestion. The revised manuscript will adjust it.

---

> > ### Comment · Reviewer_d16s · 2023-12-01
> >
> > The authors have addressed my concerns.

---

### Decision · Program_Chairs · 2023-10-24

Accept